# Building Community Driven Libraries
# of Natural Programs

**Leonardo Hernandez Cano** [* 1]  **Yewen Pu** [* 2]  **Robert Hawkins** [3]  **Josh Tenenbaum** [4]  **Armando Solar-Lezama** [1]

## Abstract

A typical way in which a machine acquires knowledge from humans is through programs – sequences of executable commands that can be composed hierarchically. By building a library of programs, a machine can quickly learn how to perform complex tasks. However, as programs are typically created for specific situations, they become brittle when the contexts change, making it difficult compound knowledge learned from different teachers and contexts. We present natural programming, a library building procedure where each program is represented as a *search problem* containing both a goal and linguistic hints on how to decompose it into sub-goals. A natural program is executed via search in a manner of hierarchical planning and guided by a large language model, effectively adapting learned programs to new contexts. After each successful execution, natural programming learns by improving search, rather than memorizing the solution sequence of commands. Simulated studies and a human experiment (n=360) on a simple crafting environment demonstrate that natural programming can robustly compose programs learned from different users and contexts, solving more complex tasks when compared to baselines that maintain libraries of command sequences.

## 1. Introduction

By compounding knowledge learned from different teachers, a student can solve increasingly complex tasks. A typical way which computers acquire knowledge from humans is through *programming* – explicit instructions on how to do the task. With the advancement of Large Language Models (LLMs), recent works have increasingly leveraged programmatic representations of policies (Liang et al., 2022; Volum et al., 2022) and planning (Huang et al., 2022; 2023; Silver et al., 2022), to take the advantage of LLMs' ability to connect programs to natural language intents. Under these formulations, a human instructs the agent through programming, either directly or with natural language, and the agent executes the program by producing a sequence of appropriate actions. Compared to learning from demonstrations, end-to-end RL, and trajectory optimizations, programmatic learning is more generalizable (Inala et al., 2020; Trivedi et al., 2021) and user-interaction efficient(Bunel et al., 2018).

A key tenant of a well learned skill is its *robustness* in the face of changing contexts. For instances: An agent operating a robot in different environments (Mesesan et al., 2019; Lin et al., 2019); A controller directing traffic in different hours of the day(Padakandla, 2021); A software operating under different dependencies(Garlan et al., 2009). As programs are explicit sequences of commands, they are *brittle* when the context changes. This brittleness accumulates in higher level programs, as failure in any sub-program results in total failure. As people tend to give context-specific instructions (Sumers et al., 2022; Bonawitz et al., 2011) (i.e. "hard coding a program" (Brown et al., 1998)), it is difficult to compound programmatic knowledge learned from different teachers and contexts (Garlan et al., 2009; Chaturvedi, 2019; Taylor, 2010; AlOmar et al., 2020).

On the other hand, humans readily generalize skills learned from different teachers and contexts. Recent works studying human to human instruction generation (Tessler et al., 2021a; Acquaviva et al., 2021; McCarthy et al., 2021b) reveal that instead of giving overtly detailed procedures, humans successfully "program" each other by providing goals and guidelines: The goal constraints the intended outcome; The guidelines increases the chances of the listener finding a solution. Work in situated actions (Suchman & Suchman, 2007) suggests that humans accomplish a task not by following an exact plan, but by improvising to adapt to the context at hand. Drawing inspirations from these concepts, we present Natural Programming (NP), a novel programming

*Equal contribution  [1]CSAIL, MIT, USA  [2]Autodesk  [3]Department of Psychology, UW-Madison, USA  [4]Department of Brain and Cognitive Sciences, MIT, USA. Correspondence to: Leonardo Hernandez Cano <leohc@mit.edu>, Yewen Pu <yewen.pu@autodesk.com>.

Interactive Learning with Implicit Human Feedback Workshop at ICML 2023

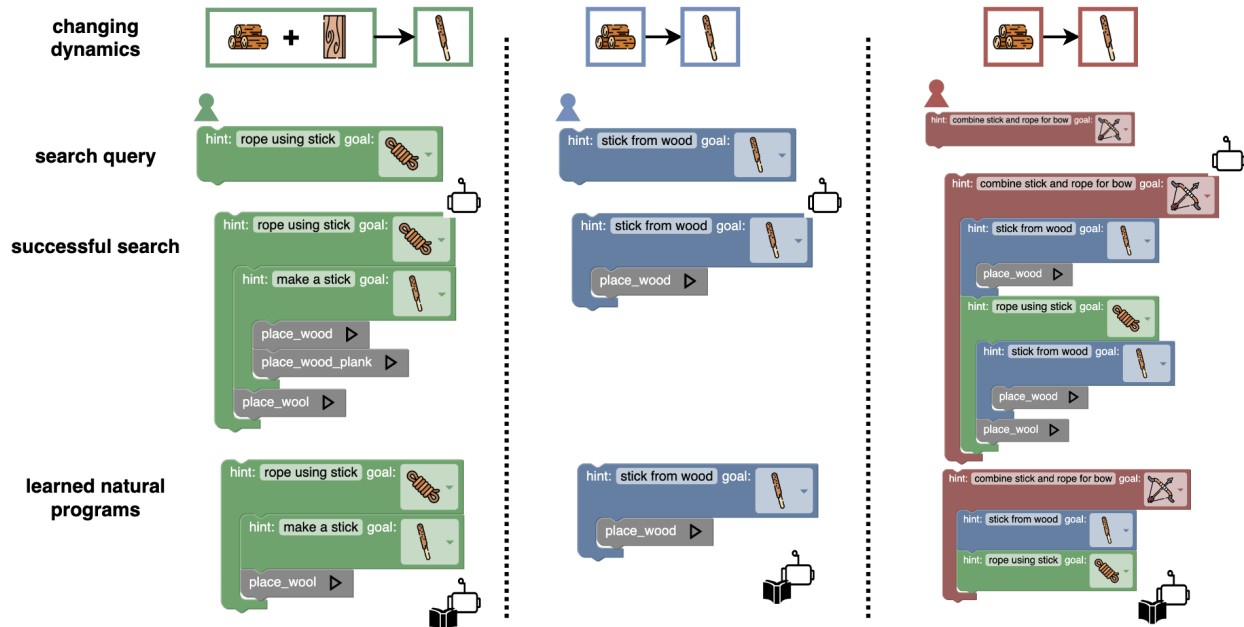

Figure 1: An overview of Natural Programming in three generations on the `CraftLite` environment, with different users and contexts. The crafting rule for a stick vary across generations, in the first generation, one needs to use a wood and a plank, while in the second and third generations, one need to use just the wood. At each generation (green, blue, red), the user programs the robot using a **search problem**, and the robot solves it via **search**, and builds a library of **natural programs** – mappings from search problems to sequences of other search problems. In the third generation, the robot is able to compose the rope program (learned from the first user) with the alternative stick program (learned from the second user) to craft a bow.

system that interacts with an user using *search problems* – consisting of a machine-interpretable $goal$[1] and a natural language $hint$. Given a search problem, NP searches for a solution in the style of hierarchical planning (Kaelbling & Lozano-Pérez, 2011; Konidaris et al., 2018; Silver et al., 2021b; Devin et al., 2017), guided by the linguistic $hint$ using a LLM. By keeping a library of search problems, NP allows itself to search for new solutions in different situations, while still benefiting from past knowledge gained elsewhere.

This work makes the following contributions:

**Formalism** – Formalizing the problem of Generational Programming under changing Contexts (GPC) to study how to create a programmatic system that can learn from different teachers and contexts.

**System** – Developing the Natural Programming (NP) system, a novel programming system that maintains a library of search problems to tackle the GPC problem.

**Evaluation** – Developing `CraftLite`, a simple yet fully fledged programming environment to study the GPC problem at scale. Evaluation of our system against realistic

baselines on both simulation and a large scale user study (n=360, type=crowd-workers, total time=90 hours) demonstrates that NP learns best from different users and contexts [2].

## 2. Related Works

We are concerned with the problem of creating a system that can learn from different teachers and contexts through programming. Programmatic policies, RL under non-stationary environments, and library learning are key aspects of our work. We highlight literature in these different areas to provide contexts for our own work, which lies in the intersection of these fields.

**Programs as Policies and Planners**   Programs are a staple representation for robot policies. Compared to end-to-end policies, programmatic policies (Andreas et al., 2017; Yang et al., 2021) are more generalizable (Inala et al., 2020; Trivedi et al., 2021), interpretable (Zhan et al., 2020; Bastani et al., 2018), and easier for humans to communicate (Bunel et al., 2018). Likewise, it is customary for planners (Shiarlis

---
[1]e.g. input-output examples, assertions, constraints

[2]We will open source the `CraftLite` environment and all the collected user-interaction dataset as well

et al., 2018; Konidaris et al., 2018; Silver et al., 2021a) to adopt a programmatic representation. With the advancement of LLMs, recent works have increasingly embraced a programmatic representation in policy (Liang et al., 2022; Volum et al., 2022) and planning (Huang et al., 2022; 2023; Silver et al., 2022), to take the advantage of LLMs' ability to translate between natural languages and programs.

**RL and Planning with Changing Dynamics** The robust operation of agents under changing dynamics (Hallak et al., 2015; Padakandla, 2021; Choi, 2000; Xie et al., 2022) is of crucial importance in realistic scenarios. Methods such as (Banerjee et al., 2017; Mesesan et al., 2019; Lin et al., 2019) calibrate a learned policy to adapt to novel environments, while (Arumugam et al., 2017; 2019; Kaelbling, 1993; Squire et al., 2015; Bodik et al., 2010) maintains a hierarchy of goal representations, allowing the agent to re-plan as environments change.

**Library Learning** In library learning, a system continually grows a library of programs, becoming more competent in solving complex tasks over time. The library learning can be self-driven (Ellis et al., 2020), guided by language (Wong et al., 2021), or in direct interactions with a community of users (Wang et al., 2017; 2016; McCarthy et al., 2021a; Karamcheti et al., 2020).

**Overall** Compared to works in robust policy and planning, we focus on *interactive learning from users* as a mean of acquiring new abstractions and search strategies. Compared to works in library learning, we focus on *goal driven task decompositions* to account for *changing contexts*. Task-oriented dialogue systems (Suhr et al., 2019; Fast et al., 2018; Wang et al., 2015) are different from our work as they do not perform library learning [3].

# 3. Generational Programming under Changing Contexts

We first define the general problem of generational learning under changing contexts, then extend it to the setting where the learning takes the form of programming.

## 3.1. Generational Learning under changing Contexts (GLC)

A GLC problem is a tuple $(\mathcal{U}, \mathcal{G}, \mathcal{S}, \mathcal{T}, \mathfrak{M})$, consisting of sequences (representing different generations) of **users** $\mathcal{U} = u_1 \dots u_n$, **goals** $\mathcal{G} = g_1 \dots g_n$, **starting states** $\mathcal{S} = s_1 \dots s_n$, **dynamics** $\mathcal{T} = t_1 \dots t_n$, and a contextual markov decision process **CMDP** $\mathfrak{M}$. This work considers the setting where the set of goals $G$ is finite and enumerable,

---

[3]i.e. these systems cannot acquire higher level programs over time

and the every dynamics $t \in T$ deterministic.

**Contextual Markov Decision Process** The CMDP (Hallak et al., 2015) maps a **context** $c = (g, s, t)$ into a specific MDP $\mathfrak{M}(g, s, t) = (S, A, s, t, R_g)$, where $S$ and $A$ are the set of **states** and **actions**, and $s, t, R_g$ are a specific **starting state**, **transition function**, and goal-conditioned **reward function** respectively. Each **generation** $i$ has a generation-specific MDP $M_i = (S, A, s_i, t_i, R_{g_i})$ where $S$ and $A$ are fixed across generations, while $s_i, t_i, R_{g_i}$ changes across generations. An MDP $M_i$ can receive a **sequence** of actions $[a_1 \dots a_k]$ and produce a final state $M_i([a_1 \dots a_k]) = s'_i$ using the transition function. The goal $g$ is a predicate function over states $g(s) \in \{0, 1\}$.

**Generational Learning** Let $\mathbf{A}[E]$ denotes a **learning agent** $\mathbf{A}$ parameterized by **experience** $E$. At each generation $i$, the agent $\mathbf{A}[E_i]$ interacts with the user $u_i$ within **environment** $\mathfrak{M}(c_i)$, which produces a **reward** and a **new experience** $(r_i, e_i) = interact(u_i, \mathbf{A}[E_i], \mathfrak{M}(c_i))$. The agent learns by growing its experience $E_{i+1} = E_i \cup \{e_i\}$, which is passed onto the next generation. The objective of GLC is to maximize cumulative reward $r = \sum_i r_i$.

## 3.2. Generational Programming under changing Contexts (GPC)

A GPC problem is a specific kind of GLC where the interaction takes place via *programming*, and the agent, a programming system, learns from experience by building a *library*. See Figure 2. Specifically, given a context $c_i$, The user constructs a **program** $p_i$. The programming system $\mathbf{A}[L_i]$, parameterized by the **library** $L_i$, *executes* the program, producing a **action sequence** and an **experience** $\mathbf{a}_i, e_i$. The sequence $\mathbf{a}_i$ is carried out on the environment $\mathfrak{M}(c_i)(\mathbf{a}_i)$, resulting in a reward if the goal $g$ is satisfied $g(\mathfrak{M}(c_i)(\mathbf{a}_i))$. The agent learns from the experience $e_i$ by building the new library $L_{i+1}$. The programming systems in a GPC differ by their respective forms of programs, how they execute these programs, and how they maintain and grow their libraries – the crux of this work.

# 4. Programming Systems

We describe two baseline programming systems, direct programming and direct synthesis, that keep libraries of functions. For each, we explain the forms of programs they require, how they execute a program to produce an action sequence, and how they grow libraries across generations.

## 4.1. Direct Programming (DP)

DP emulates library building in traditional programming, where a library of functions can be accessed and expanded by a community of users. For this work, we consider a simple family programs without variables and control flows

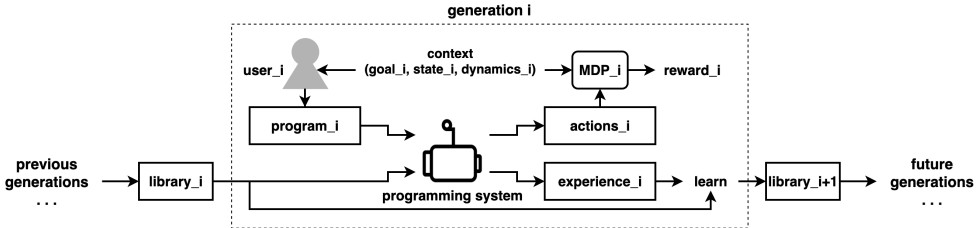

Figure 2: The GPC (generational programming under changing context) problem at a particular generation. Given a particular context, the user writes a program. This program is then executed by the agent (a programming system) producing a sequence of actions, which is then carried out on the contextual MDP, resulting in a reward. The agent also generates an experience, allowing it to build a better library for future generations.

[4]. The library $L$ is a mapping from a function name $f.name$ – a string, to a function body $f.body$ – a sequence of actions. $L[name] = [a_1 \ldots a_k], a_i \in A$. The user $u$ produces a program $p$ by reasoning over the context $c$ and the library $L$. $u : (c, L) \to p$. The program $p$ can take two forms: (1) executing an existing function; (2) defining a new function.

**Execution** The program is a single string $name$, which the user selects from the library $L$, and the corresponding action sequence $L[name]$ is produced. No experience is produced by the system.

**Function Definition** The user gives DP a mapping $name_{new} \to [name_1 \ldots name_k]$. The system produces a concatenated action sequence $\mathbf{a}_{new} = L[name_1] + \cdots + L[name_k]$. The system also produces an experience $e = name_{new} \to \mathbf{a}_{new}$.

**Generational Library Learning** DP starts with the library of primitives $L_1 = L_A = \{ \text{"a1"} \to [a_1] \ldots \text{"an"} \to [a_n] \, \forall a_i \in A \}$, where all the actions in $A$ maybe referred to by their names. DP grows the library as user adds new functions to it $L_{i+1} = L_i \cup \{name_{new} \to \mathbf{a}_{new}\}$.

### 4.2. Direct Synthesis (DS)

DS augments DP with language-guided program synthesis, where the system searches the library for a satisfying sequence. DS maintains a library of functions $L$ same as in DP, and in addition, it is made aware of the set of goals $G$ and can query the environment $\mathfrak{M}(c)$ *as a black box*[5].

**Programs are Search Problems** The user $u$ programs DS by querying it with a **search problem** $q = (g, hint) \in Q$, where $Q = G \times NL$, a cross product of all goals and all natural language strings. Specifically, $u : (g, s, t) \to (g, hint)$. Instead of browsing the library $L$, the user specifies the goal $g$ to DS directly, along with a $hint$ on how to achieve it.

**Execution via Search** DS searches for an action sequence $\mathbf{a}$ such that, once carried out in $\mathfrak{M}(c)$, satisfies the goal $g$. In a typical program synthesis fashion (Devin et al., 2017; Ellis et al., 2020), it performs rejection sampling[6]:

Algorithm 1: DS Execution
```
DS((g, hint), M(c), L) =
  for L^i in L^1, L^2, ..., L^max_len:
    repeat n_iter times:
      f_1 ... f_i ~ propose(f_1 ... f_i ∈ L^i|(g,hint), L^i)
      a = f_1.body + ... + f_i.body
      if g(M(c)(a)):
        return a
  return Fail.
```

The search is incremental, where DS attempts to first find a single program $f \in L$ that satisfies the goal $g$, and when that fails, expands the search to $[f_1, f_2] \in L^2$, and so on.

**Propose** The rejection sampling is guided by a *propose* function in the manner of language-guided synthesis(Wong et al., 2021; Li et al., 2022), where depending on the linguistic hint, certain combinations of functions become more likely to be sampled. This is crucial as the search space of $L^i$ grows exponentially as $i$ increases. The difficulty of having a good propose function is *a lack of data* – prior approaches such as (Wong et al., 2021; Li et al., 2022; Suhr et al., 2019; Wang et al., 2015) required significant labeling efforts of paired instances of natural-language to programs (in our case, function sequences). Instead, we look to pretrained LLMs as a source of prior knowledge. Specifically, we consider two "backends" of our propose function, one based on semantic similarities, the other based on prompting:

$$propose_{sim}(f_1 \ldots f_i | (g, hint), L^i)$$
$$\propto \|embed_{LLM}(f_1.name + \cdots + f_i.name)$$
$$- embed_{LLM}(hint)\|$$
$$propose_{prompt}(f_1 \ldots f_i | (g, hint), L^i)$$
$$= \texttt{LLM(prompt[hint,L])}$$

$propose_{sim}$ ranks the function composition $f_1 \ldots f_i$ based

---

[4]think of a system of macros

[5]i.e. the dynamics of $\mathfrak{M}(c)$ hidden from the agent, but it can take actions and observe outcomes

[6]most program synthesis approaches adopt the generate and check framework

on how similar their concatenated strings are is to the $hint$, based on their embedded distances under a LLM. $propose_{prompt}$ simply gives an instruction tuned LLM with the hint and a subset of the library's content, and queries the LLM for a sequence of function names. Keep in mind that it is not the contribution of this work to find *the best* implementation for the $propose$ function, but rather, we highlight that our approaches are compatible with standard techniques.

**Generational Library Learning** DS starts with a library of primitives $L_1 = L_A$. Given a search problem $q = (g, hint)$, DS only produces a sequence $\mathbf{a}$ when $g(\mathfrak{M}(c)(\mathbf{a}))$, and $e = hint \to \mathbf{a}$ is added to the library. Doing so achieves *compression* (Ellis et al., 2020) – making a long sequence of functions (that is difficult to search for) to a single function.

### 4.3. Brittleness of Functions

Consider a particular sequence $\mathbf{a}_1$ made under a particular CMDP $\mathfrak{M}(c_1)$ which satisfies the goal $g$. While sequence $\mathbf{a}_1$ satisfies the goal $g(\mathfrak{M}(c_1)(\mathbf{a}_1))$, the same sequence is unlikely to satisfy the same goal under a different context $c_2$, i.e. $g(\mathfrak{M}(c_1)(\mathbf{a}_1)) \neq g(\mathfrak{M}(c_2)(\mathbf{a}_1))$. As a result, when contexts change, much of the functions in the library $L$ can no longer satisfy the goals they are written for.

## 5. Natural Programming

Notice the discrepancy in the DS system: it takes in search problems, yet maintains a library of a different kind – a library of explicit action sequences. What if we maintain a library of search problems as well? A **natural program** $np \in \mathcal{P}$ is a mapping from a search problem to a sequence of primitives or other search problems. $\mathcal{P} = \{Q \Rightarrow (L_A \cup Q) \times \cdots \times (L_A \cup Q)\}$. The Natural Programming system maintains a library of natural programs $\mathcal{L} \subset \mathcal{P}$.

**Programs are Search Problems** The interface of DS and NP are exactly the same – given a context, an user produces a search problem $q \in Q$ for the system.

**Execution via Recursive Decomposition** Given $q = (g, hint)$, NP searches for a satisfying action sequence in the manner of hierarchical planning similar to the MAXQ (Dietterich, 2000) algorithm. For simplicity, we have NP only returning a satisfying final state $s'$ or $Fail$ if it cannot finds it – the satisfying action sequence $\mathbf{a}$ can be recovered by instrumenting book-keeping variables. We also use $\mathfrak{M}(c)(s, a)$ to denote taking an action step $a$ with a starting state $s$ on $\mathfrak{M}(c)$.

Algorithm 2: NP Execution

```
NP(s, q, L, M(c)):
  case q is Action:
    return M(c)(s,a)
```

```
case q is (g, hint):
  decompose = Queue([propose(q,L)...n_times])
  qs_i = decompose.pop() (label: A)
  cur_state = s
  for qs_i_j in qs_i:
    nxt_state = NP(
        cur_state, qs_i_j, L, M(c)
    )
    if next_state is not Fail:
      cur_state = nxt_state
    else:
      goto (A)
  if q.goal(cur_state):
    return cur_state
  else if decompose is empty:
    return Fail.
  else:
      goto (A)
```

The most salient aspect of the NP execution is the recursive call to NP itself[7]. Thus, unlike DS, NP can keep decomposing until a satisfying sequence is found. Both DS and NP share the same $propose$ implementation. In practice, we implement Algorithm 2 with a priority queue and caching to avoid redundant executions.

**Generational Library Learning** When Algorithm 2 finds a satisfying $\mathbf{a}$, NP produces an experience in the form of a *search tree* (Figure 1, middle row), consisting of the correctly chosen decomposition steps. Each correct decomposition is a mapping $q \to q_1 \ldots q_j \in \mathcal{P}$. They are added to $\mathcal{L}$ (Figure 1).

## 6. CraftLite

To study the effects of different programming systems in the context of GPC, we introduce a simple programming environment, CraftLite, with the following desiderata: (1) Can be used as a multi-generational GPC problem. (2) Crowd workers can consistently learn how to "program" it in under 5 minutes – so a large scale user study would not be prohibitively expensive. (3) Has a real-time ($\sim$2s) responsiveness to accommodate for end-user interactions.

### 6.1. CraftLite as a Conditional MDP

**State** A state consists of an inventory of multiple items, two crafting input slots, and a single crafting output. There 29 total possible items in CraftLite.

**Action** There are only 2 kinds of **actions**, input_x where x is an item name, and craft. input_x moves an existing item in the inventory to the input slots, and craft moves the transformed item in the output slot back to the inventory, consuming the inputs. Thus, there are 30 total actions.

---

[7]the goto is only there for ease of explanation!

**Dynamics** The dynamic is dictated by the `recipe` book, which dictates which two input items can be successfully transformed into an output item. Out of the 29 items, 4 are "raw materials", and 25 are "craftable items" with 2 possible crafting rules: For instance, a stick is made with rock+plank in one version while wood+plank in another. A recipe book can be *randomly generated* by choosing one version of the crafting rules – there are $2^{25}$ possible recipe books total. Depending on the particular recipe book, the most complex item can take up to 87 action steps. Thus, the raw command sequence complexity of `CraftLite` is $30^{87}$.

**Goal** is a list of `goal items`, which generates a reward when it is first added to the inventory.

## 6.2. The GPC Problem using **CraftLite**

The GPC set up for `CraftLite` is slightly different from that defined in Section 3 to allow for an user to interact with a programming system for an extended period of time in the same environment.

**Session** A **session** $g_i$ at generation $i$ consists of: an user $u_i$, a programming system with library $\mathbf{A}[L_i]$, and a context $c_i$. The **context** $c_i = (g_i, s_0, t_i)$ consists of: a list of goal items $g_i = g_i^1 \ldots g_i^r$, the starting state $s_i$, and a randomly generated recipe book $t_i$. At the start of the interaction, a CMDP $\mathfrak{M}(c_i)$ is created. Within a specified time limit (e.g., 10 minutes), a user $u_i$ interacts with $\mathbf{A}[L_i]$ to craft as many goal items as possible under the same environment $\mathfrak{M}(c_i)$.

**Chains** The participants are organized into "cultural chains"(Tamariz & Kirby, 2016; Tessler et al., 2021b) collaborating with the same programming system. A chain is a sequence sessions: $(u_1, \mathbf{A}, c_1), \ldots (u_n, \mathbf{A}, c_n)$. The programming system starts with the initial library $L_1$. This library is updated after every interaction with an user, and persists to the next generation.

**Batch** A batch consists of multiple chains (one for each programming system) with a paired sequence of contexts. For instance, the following is a batch of 2 conditions $\mathbf{A}_1, \mathbf{A}_2$, $2n$ participants, and paired contexts $c_1 \ldots c_n$:

$$(u_1, \mathbf{A}_1, c_1), \ldots, (u_n, \mathbf{A}_1, c_n);$$
$$(u_{n+1}, \mathbf{A}_2, c_1), \ldots, (u_{n+n}, \mathbf{A}_2, c_n);$$

## 6.3. User Interface of **CraftLite**

The user interface of `CraftLite` consists of a left programmatic panel based on Blockly (Pasternak et al., 2017), and a right panel showing the current state. See Figure 3

## 7. Experiments

We seek to answer the following research questions. **RQ1**: Is NP effective at solving the GPC problems in `CraftLite`? **RQ2**: Does the capability of NP improve at a faster rate across generations? **RQ3**: Does NP allows the user to achieve more tasks with fewer efforts?

## 7.1. Simulated Study

We conduct a simulation study with a large number of simulated, idealized users to study the relative effectiveness of NP vs DS as a function of changing dynamics.

**Controlling Dynamics** In `CraftLite`, every craftable item has 2 possible rules. We can control how much dynamics can vary from one generation to next by adjusting the probability $r$, of how likely the second rule is chosen. A $r = 0$ would cause the generated recipe to only contain the first rule for every item, i.e. *no* dynamic changes across generations, while a $r = 0.5$ will uniformly sample one of the $2^{25}$ possible recipe books – the most difficult setup where dynamics changes most.

**Batches** At every generation, the context contains: (1) A set of 6 randomly chosen "leaf items" were used as goals – craftable items that are not required to craft other items. (2) The starting state consisting of 4 raw materials. (3) A randomly generated recipe with a particular $r$ value. A batch consists of 20 generations of 2 conditions, DS vs NP, where the same generation shares the context.

$$(u_{sim}, \text{DS}, c_1), \ldots, (u_{sim}, \text{DS}, c_{20});$$
$$(u_{sim}, \text{NP}, c_1), \ldots, (u_{sim}, \text{NP}, c_{20})$$

For each 3 value of $r = [0, 0.25, 0.5]$, we generate 5 batches each.

**Simulated Users** The simulated user is idealized. It will attempt first to craft a given goal item, and wait some amount of time until the solver succeeds or fails. If the solver succeeds, it moves onto the next item. If the solver fails, it recursively attempts to craft a pre-requisite item. We do not simulate an user for DP, and defer evaluation of DP to a real human experiment.

**Session** Each session has a total of 2 minutes time limit, and the solver timeout after 10 seconds.

**Results** **RQ1**: As we can see, when dynamics are kept constant ($r = 0.0$), DS and NP performs identically on `CraftLite`. However, as dynamics vary more ($r = 0.25$, $r = 0.50$), the performance of NP is superior to that of DS (Figure 4). **RQ2**: Similarly, we find NP improves at a faster rate across generations compared to DS under more dynamics variations. **RQ3**: Because these are simulated users, we defer the measurement of user efforts to the more realistic scenario of a human experiment.

**Propose Back-end** We found that for `CraftLite`, even if we give a lenient solver timeout of 10 seconds, enough for a

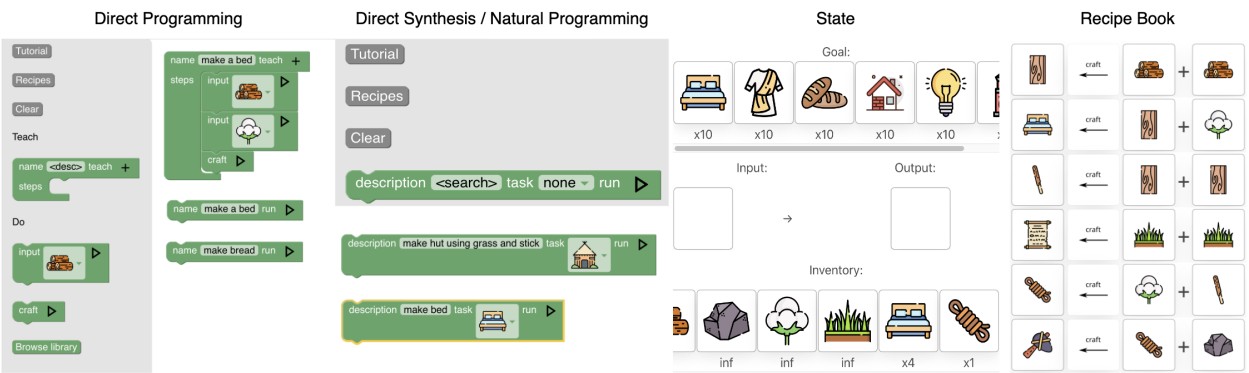

Figure 3: The `CraftLite` UI. The DP programming system allows the user to manually define new functions, and browse a library of existing functions. The DS / NP system only allows user to give a search problem. The game state shows an inventory of current items, and a list of goal items to be completed. All systems have a recipe book the user can browse, which encodes the dynamics.

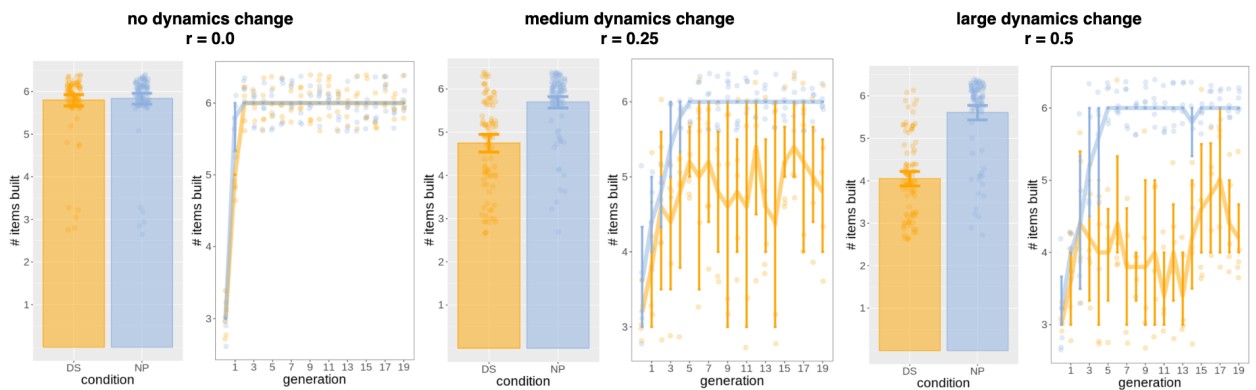

Figure 4: Simulation on how different programming systems perform under different amount of dynamic changes across generations. Total of 600 idealized simulated users. $r = 0.0$ no dynamic change and $r = 0.5$ most dynamic change. Error bars are 95% CIs (nboot=1000), dots represent individual sessions outcomes.

prompting back-end to give a response, the similarity based propose function is always faster. A comparison of semantic-distance vs prompting can be found in the supplement.

## 7.2. Human Experiment

**Procedure**  We recruited 360 participants from the Prolific crowdsourcing platform. Participants were recruited from the U.K. and the U.S., excluding people without English proficiency. We paid an average of $12.06 USD per hour, including bonuses for $0.4 USD per "goal". In total, we ran 360 sessions across 12 batches of paired contexts, and of a length of 12 generations. 6 batches containing all three conditions (DP, DS, NP) and 6 only containing the two of greatest interest (DS, NP). Each session is 10 minutes. The DS and NP solver is set to have a maximum timeout of 30

seconds, but the user can cancel the solver at any point.

### 7.2.1. RESULTS

**RQ1: More items are built overall with NP**  Across all batches and generations, we find that NP allows people to build the most goal items overall (Figure 5A; DP mean = 1.7 items, $95\%CI = [1.6, 1.9]$, DS mean = 2.4 items, $95\%CI = [2.3, 2.6]$, NP mean = 2.9 items, $95\%CI = [2.7, 3.1]$). Both synthesis-based systems perform significantly better overall than direct programming ($t(71) = 5.6, p < 0.001$ for DP vs. DS; $t(71) = 7.9, p < 0.001$ for DP vs. NP, paired). Compared to the DS, NP produced significantly more items ($t(143) = 4.7, p < 0.001$).

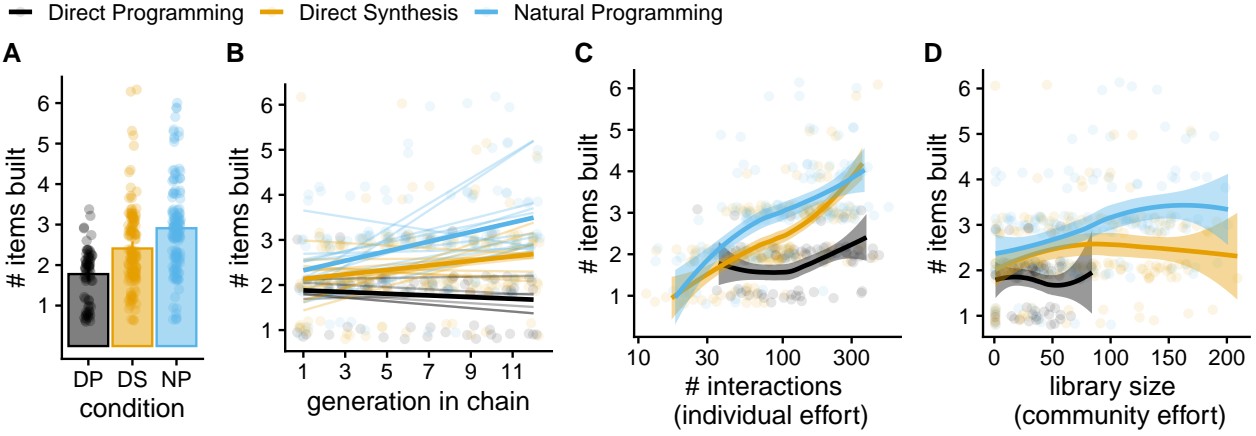

Figure 5: Natural programming (NP) enables (A) more items to be build overall, (B) improves significantly as successive generations interact with the system, and (C-D) reduces the effort required to reach the same performance. Error bars are 95% CIs; low-transparency dots represent individual sessions; low-transparency lines represent regression fits for individual chains.

**RQ2: NP chains improve more rapidly** Next, we consider how NP improves as additional users in a chain interact with the system (Figure 5B). We fit a (Bayesian) mixed-effects linear regression model predicting the number of items built as a function of the generation in the chain (integer from 1 to 12), the programming system (categorical, NP vs. DS vs. DP), and corresponding interaction terms. First, examining the NP condition alone, we found that performance improved significantly across generations, ($b = 6.24, 95\%$ credible interval $= [3.3, 9.3]$). We found that this slope was meaningfully larger than the DS condition (diff $= +3.4, 95\%$ credible interval $= [-0.5, 7.1]$) and the DP condition (diff $= 8.6, 95\%$ credible interval $= [3.5, 13.5]$).

**RQ3: NP requires less effort** What properties of NP enable these performance benefits? We know it is not due to the difference of user interface, as DS and NP uses exactly the same interface. In this section, we argue instead that NP reduces the amount of effort required to obtain the same results. We tested this effect by running Bayesian mixed-effects regressions predicting the number of items built as a function of effort and the programming system being used. We consider two different metrics of effort. First, we examine *individual effort*, the (log) number of "submissions" – registered every time a user attempts to execute a program – made by a user to the programming system (Figure 5C). At a given level of effort, we found that participants in the NP condition were able to craft significantly more items than those in DP condition (diff $= 1.25$ items, $95\%CI = [0.8, 1.7]$) or the DS condition (diff $= 0.42$ items,

$95\%CI = [0.2, 0.6]$). Concretely, among participants that crafted exactly 2 items, DP required an average of 156 submissions, DS required 81 submissions, and NP only required 63 submissions. Similar results were found for *collective effort* – the total library size (i.e., unique and successful previous interactions) that has accumulated at the given point in the chain (Figure 5D).

## 8. Conclusion

We define a new class of problems, GPC, to study how an agent can learn from different teachers and contexts programmatically. We developed the NP system – based on a hierarchical planner that learns new abstractions and decomposition rules from users. We demonstrate NP is efficient in solving GPC problems in the `CraftLite` domain through both simulations and a large (n=360) user study.

**Limitations** The immediate limitations of this work is `CraftLite` is rudimentary in its *complexity*. Scaling up NP requires better *propose* implementations – for instance, a faster LLM+prompting, and taking in current state as context. Scaling up GPC requires a richer task domain than `CraftLite`, yet still making it intuitive for end-users for a systematic evaluation. Ultimately, a richer task domain will cost more money. The more subtle limitation of this work is the perfect mutual understanding of goals between user to agents (DS and NP). This is possible in `CraftLite`, but more generally, this is the complex topic of AI-alignment(Gabriel, 2020). How to robustly compose knowledge when under imperfect mutual understandings of goals is an exciting field for future research.

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
