# OpenReview forum: "Building Community Driven Libraries of Natural Programs"
_ICML.cc/2023/Workshop/ILHF — ILHF Workshop ICML 2023_

### Official Review · Reviewer_L2wC · 2023-06-19
**A nice perspective on alternate implicit feedback mechanisms, unclear the generalizability**

**Rating:** 6
**Confidence:** 3

**Review:**

This paper presents Natural Programming (NP), a programming paradigm that supports interaction with an end user via search supported by an LLM. The authors define a Generational Learning under Context problem setup where CMDPs are generated for each reward/user/environment. Specifically, the authors focus on an interaction via learning a library of programming abstractions.

Strengths:
- a clear and formal definition of an interactive natural programming problem under context
- a nice perspective on an alternate form of implicit human feedback, i.e. natural programs
- a neat (albeit intuitive) way of leveraging LLMs to help guide search through the program space

Weaknesses:
- unclear which problems in practice actually exhibit the type of "search and index-able" programs required by the algorithm
- the writing post literature review is increasingly unclear - what exactly is the proposed contribution independent from existing natural programming setups? It's difficult to disentangle what is the class of problems that's proposed vs. the actual algorithm that is intended to solve them.
- the proposed domain, CraftLite, is pretty simple--which in itself is not a huge problem, but I don't see any intuition re: how this would extend to more complex environments where actions and states are not discrete

---

### Decision · Program_Chairs · 2023-06-20

Accept